# Comparative Study on the Hardness and Wear Resistance of the Remelted Gradient Layer on Ductile Iron Fabricated by Plasma Transferred Arc

**Botao Xiao [1],\*, Xuefang Yan [1], Wenming Jiang [2], Zitian Fan [2], Qiwen Huang [3], Jun Fang [1] and Junhuai Xiang [1],\***

[1] Jiangxi Key Laboratory of Surface Engineering, Jiangxi Science and Technology Normal University, Nanchang 330013, China; yan1441971268@hotmail.com (X.Y.); fangjun@jxstnu.edu.cn (J.F.)
[2] State Key Laboratory of Material Processing and Die & Mould Technology, Huazhong University of Science & Technology, Wuhan 430074, China; wmjiang@hust.edu.cn (W.J.); fanzt@hust.edu.cn (Z.F.)
[3] Wuhan Huacai Surface Technology Co., Ltd., Wuhan 430074, China; hmst66@hotmail.com
\* Correspondence: 1020130973@jxstnu.edu.cn (B.X.); 1020101011@jxstnu.edu.cn (J.X.)

**Abstract:** Repairing the worn surfaces of wear-resistant workpieces, such as rollers, is one of the main application fields of surface treatment, but the repairing time is often not considered. In fact, the repairing time is very important, since it affects the repair quality and service life of wear-resistant workpieces. In this paper, a remelted gradient coating was prepared on a ductile iron plate by plasma transferred arc to simulate the repair treatment of wear-resistant workpieces. First, two positions in the remelted gradient coating were defined, i.e., the top of the gradient remelted layer was defined as M1, and the position where the hardness was two-thirds of the top of the remelting layer was defined as M2. Next, the time taken to repair the workpiece when the working surface reached M2 was proposed. Finally this method was verified by a comparative study on the microhardness and wear resistance of the M1 and M2. In this paper, the M2 was located at a ~0.5 mm from the top of the remelted gradient layer. The results show that the microhardness of the position of the M1 was higher than that of the position of the M2. However, the wear resistance of the M1 was worse, as confirmed by the wear rates. At the same time, cracks and fragments were observed on the worn surface of the M1 and M2 positions. Furthermore, the coefficient of friction (COF) of the position of M1 was noted to be first higher and subsequently lower than that of the position of M2, owing to the grinding ball entering the substrate. The abrasion mechanisms of both regions were observed to be complex, including oxidative wear, adhesive wear, delamination wear, and/or fretting wear. The experimental data indicate that it is feasible to determine the repair time according to the microhardness of workpieces.

**Keywords:** microhardness; wear resistance; remelting; ductile iron; plasma transferred arc

## 1. Introduction

Serious wear issues in workpieces made with ductile iron, such as rollers, industrial valves, etc., have led to an urgent demand for improvements in hardness and wear resistance. Therefore the worn surfaces of these workpieces are repaired in the production process. This is an economical and simple method that is of great significance in increasing the service life of workpieces and reducing energy consumption in the industry. As far as the production technology is concerned, the material modification techniques usually used in the industry include casting [1–3], forging [4–6], welding [7,8], heat treatment [9–11], and surface modification [12–16]. Among these techniques, research into the use of high-energy beams to improve the microstructure and performance workpiece surfaces has received significant attention because high-energy beams have a wide range of potential applications in workpiece-surface modification, since they offer significant advantages during repair

of these worn surfaces, including high energy density and processing speed. Ductile iron usually serves as a workpiece material, and its performance needs to be further improved to fulfil the demands of the rapid development of industrial technology. Many literature reports indicate that the microstructure and properties of ductile iron were improved using a high-energy-density arc with laser and plasma [17–25]. Li et al. [20] simulated the remanufacture processing of ductile iron using laser cladding to improve the microstructure and mechanical properties by using different groove and cross-cladding processes; the results indicated that the microstructure of the surface cladding was martensite and lamellar cementite, that the microhardness decreased significantly in the bottom of the grooves, and that the ultimate strength of the specimen was 502 MPa. Cheng et al. [21] comparatively analyzed the surface-melted microstructure and properties of gray and nodular cast irons by using a plasma transferred arc; the graphite dissolved near the surface and a white cast-iron structure transformed after the solidification. After solidification, the microhardness in the surface layer was higher than that in the specimen substrate. Ceschini et al. [22] first created a surface treatment for ductile iron at two different energy intensities and then carried out dry sliding wear. The results showed the lower depth of the hardened layer, and the surface hardness was measured in the ductile iron treated by the lower-energy-intensity laser, while the wear resistance of the ductile iron was enhanced. In addition to high-energy-beam surface remelting, the surface cladding and surface alloying of ductile iron have also been studied. Pagano et al. [23] studied the transformation of the microstructures, hardness, and tribological performance of a ductile cast-iron surface remelted by laser, and the results indicated that the wear resistance, hardness, and coefficient of friction of the ferritic ductile cast iron were increased simultaneously. Cao et al. [24,25] remelted and alloyed on the surface of ductile iron using a plasma transferred arc. The results showed that either remelting or alloying can improve the surface hardness and wear resistance of ductile iron.

It can be seen from the aforementioned analysis that surface modification is an effective way to improve the surface microstructure and performance of workpieces made with ductile iron, especially their hardness and wear resistance, but the repair time of workpieces made with ductile iron has not received attention from researchers. In fact, to determine the repairing time of workpieces made with ductile iron is crucial for industries because it influences the quality and service life of workpieces.

In this study, the top of the gradient remelted layer was defined as M1, and the position where the hardness was two-thirds of the top of the remelting layer was defined as M2. The aim was to determine the repairing time of workpieces made with ductile iron according to a surface hardness of two-thirds, i.e., after the workpiece was repaired, a portable hardness tester was used to measure the hardness of the surface. When the hardness was two-thirds of the surface hardness, it was necessary to repair the worn surface again. Finally, the feasibility of this method was verified by the comparative microstructure, microhardness, and wear resistance of the M1 and M2 positions in the remelted gradient layer fabricated by the plasma transferred arc. This study will provide a theoretical foundation for determining the repair time of the workpieces made by the ductile iron.

## 2. Materials and Methods

The substrate used was EN-GJS-500-7 (3.42 wt% C, 2.58 wt% Si, 0.38 wt% Mn, 0.014 wt% S, and 0.040 wt% P). The substrate's microstructure included ferrite, pearlite and graphite nodules, as shown in Figure 1.

The surface remelting of the ductile iron was carried out at the plasma transferred-arc surface-modification processing center (developed by Wuhan Huacai Surface Technology Company, Wuhan, Hubei, China) [24]. A diagram of plasma processing is shown in Figure 2a.

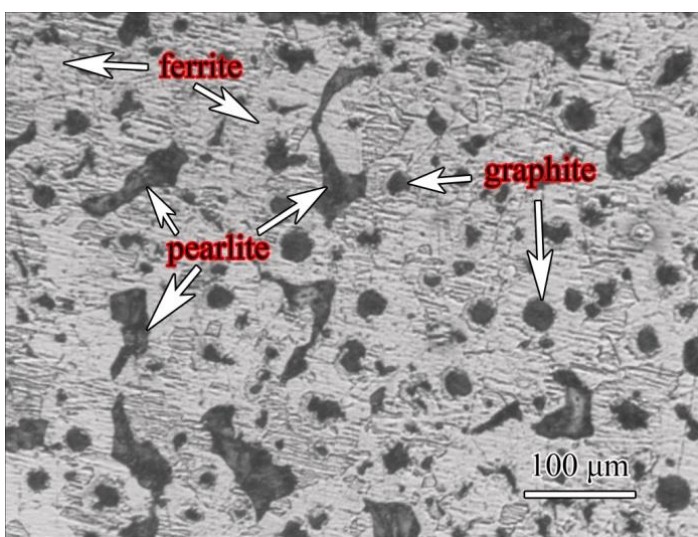

**Figure 1.** Optical micrograph demonstrating the microstructure of the substrate material.

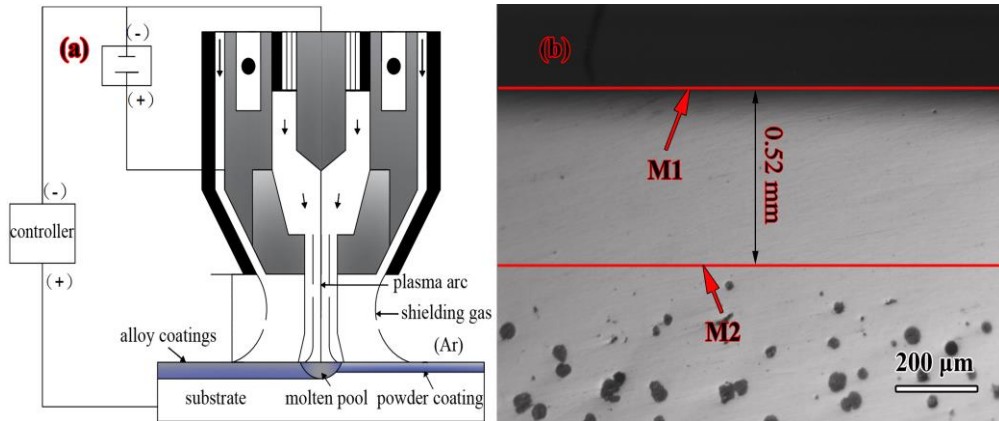

**Figure 2.** Schematic diagram of remelting treatment and determined research position: (**a**) the processing principle of plasma transferred arc, (**b**) the position definitions of M1 and M2 in this paper.

The processing parameters were selected after several optimization experiments, which were as follows: arc current of 60 A, scanning speed of 1000 mm/min, nozzle diameter of 4 mm of plasma transferred arc, 4 mm of work distance, 2 L/min of plasma gas flux (Ar), and an overlap ratio between tracks of 25%. After the surface remelted, the cross-section specimens were sectioned, ground, and polished, and then the microhardness was measured using a TMVS-1 microhardness tester with a load of 1.96 N and a dwell time of 15 s to confirm the position of the M2, which was located about 0.52 mm from the top of the remelted gradient layer, as shown in Figure 2b. Next, the specimens were etched by a solution of 4 mL nitric acid and 96 mL ethanol to reveal the microstructure in the remelted gradient layer. The microstructure was analyzed using XJL-03 optical microscope (OM) and scanning electron microscope (Leo 1530 FEG SEM, Carl Zeiss AG Co., Ltd., Oberkochen, Germany) in combination with energy-dispersive X-ray spectroscopy (INCA EDS, Oxford Instruments Co., Ltd., Bucks, England) on the cross-sectional specimens. The phase composition in the M1 and M2 was investigated using X-ray diffraction (XRD-6100, Shimadzu Instruments Co., Ltd., Kyoto, Japan) with Cu-K$\alpha$ radiation generated at 40 kV and 40 mA, with a scanning speed of 5°/min. Subsequently, the layers with thicknesses of 0.52 mm from the top of remelted gradient layer were first ground and then polished. Finally, the wear tests were conducted with a ball-on-disk apparatus (MFT-R4000 tester, Lanzhou Huahui Instrument Technology Company, Lanzhou, China) under dry sliding conditions at room temperature. The disc was made of the M1 and M2, while the counter-

part ball, which had a diameter of 6.3 mm, was composed of $Si_3N_4$ ceramic. The wear tests were carried out at a normal load of 20 N and a sliding speed of 20 mm/s, for a duration of 30 min. The coefficient of friction (COF) generated between the contacting ball and disc was measured automatically and continuously recorded during the test. Subsequently, the worn surfaces of the specimens were observed using scanning electron microscope (SEM). The weight loss from the discs was measured using a precision balance, with an accuracy of $\pm0.0001$ g. Subsequently, the wear rate ($\omega$) of the specimens was calculated using Equation (1) [26]:

$$\omega = w/SF \tag{1}$$

where $w$ is the wear loss in grams, $S$ is the total sliding distance in meters, and $F$ represents the load in the wear tests. Three repeat tests were performed for each frictional pair, and the reported values represent the average of the three repeat tests.

## 3. Results and Discussion

### 3.1. Microstructural Characteristics and Microhardness

The graphite morphologies from the remelted gradient zone (RZ), heat-affected zone (HZ), and substrate [24,25] are shown in Figure 3. As can be seen from Figure 3a, the graphite nodule in the remelted gradient zone disappeared. As the depth from the surface of the remelted gradient layer increased, the graphite's morphology was observed to be similar to that in the substrate. As the remelted gradient layer formed during the processing of the ductile iron, the graphite nodule in the remelted gradient layer fused under a high energy input, followed by its disappearance in the remelted gradient zone. This phenomenon was similar to the findings reported in the literature [20]. The observed phenomenon results were due to the short remelting time owing to the fast scanning speed of the plasma transferred arc, which resulted in the quick solidification of the remelted gradient layer associated with self-quenching. Furthermore, the inoculant element resulting in ductile iron due to the high temperature induced by the plasma transferred arc and the nucleoid core of the graphite nodule disappeared in the remelted gradient layer.

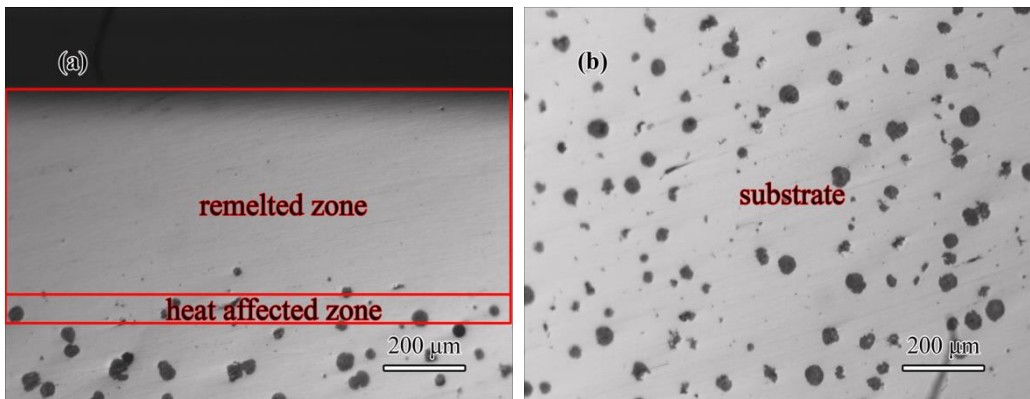

**Figure 3.** The graphite morphology in the remelted gradient layer fabricated by the plasma transferred arc: (**a**) remelted gradient layer and (**b**) substrate.

Figure 4 demonstrates the microstructure of the ductile iron gradient remelted by the plasma transferred arc, and the remelted-gradient and heat-affected zones were observed to form on the surface of the ductile iron.

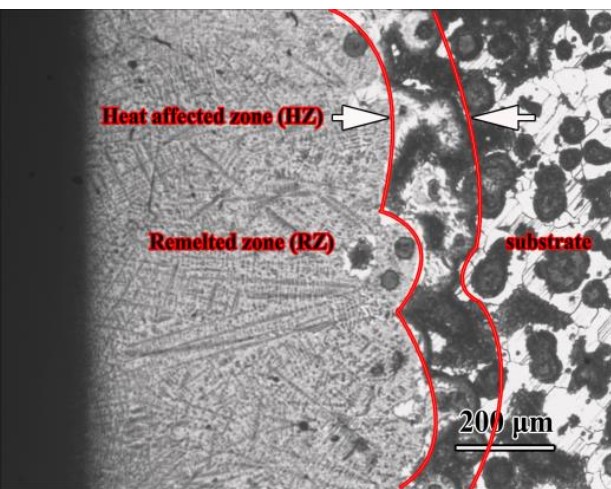

**Figure 4.** The microstructure of ductile iron gradient remelted by the plasma transferred arc includes remelted zone (RZ), heat-affected zone (HZ), and substrate.

As can be seen from Figure 4, the microstructure of the remelted gradient layer presented the characteristics of hypoeutectic white cast iron. Similar to previous literature reports [21], the microstructure of the remelted gradient zone transformed from the ferrite, pearlite, and graphite nodules to the residual austenite and ledeburite dendrites. Meanwhile, the heat-affected zone was observed to form between the remelted gradient zone and the substrate. This was mainly related to the temperature gradient formed during the cooling of the remelted gradient layer. Two methods of heat release could be used for the solidification of the gradient remelted layer: one way was to release the heat to air, and the other involved its absorbtion by the substrate. The cooling rate of the remelted gradient layer was fast, resulting in a temperature gradient, which promoted the formation of the hypoeutectic white cast iron. Under the action of the plasma transferred arc, the remelted gradient temperature rapidly increased, and a fraction of the melting carbon atoms dissolved in the remelted gradient layer. However, most of the melting carbon atoms did not dissolve in the remelted gradient layer as the solubility of the carbon in the $\alpha$-iron was low. In addition, the solubility of the carbon in the iron liquids reduced in decreasing temperatures in the remelted gradient layer. As the remelted gradient layer solidified, the nucleation of the carbon atom could not occur and the graphite nodule disappeared in the remelted gradient layer. This led to the precipitation of the carbon atoms from the remelted gradient layer. At the same time, the carbon atoms that melted into the remelted gradient layer were continuously precipitated and dissolved during the solidification of the remelted gradient layer under argon flow.

A SEM analysis was carried out to gain further insights into the microstructure of the remelted gradient layer, and the images of the field-emission scanning electron microscope (FISEM) are shown in Figure 4. The microstructure demonstrated that the fine grains on the surface of the remelted gradient layer consisted of residual austenite and cementite, as shown in Figure 5a,b.

It was confirmed that the self-quenching restrained the reformation of the graphite nodule. On enhancing the depth from the top of the remelted gradient layer, the extent of needle-like cementite reduced, while the amount of retained austenite was larger compared to the M1. It is a well-known fact that microstructures with needle-like cementite can improve wear resistance.

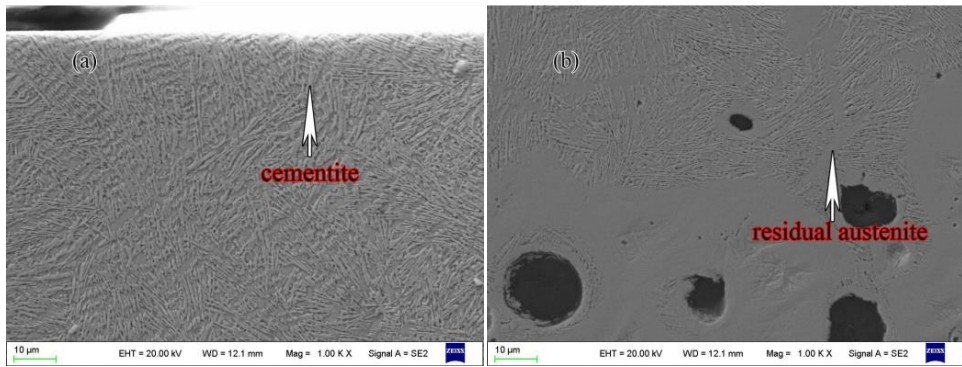

**Figure 5.** FISEM morphology of ductile iron treated by the plasma transferred arc: (**a**) remelted gradient layer and (**b**) heat-affected zone and substrate.

The microhardness was measured along a cross-section extending over a depth of approximately 0.7 mm from the M1, as shown in Figure 6. The results indicated that the microhardness peaks were 871 HV0.2 at the M1 and decreased towards the substrate. The higher microhardness of the M1 can be attributed to the presence of fine cementite and abundant ledeburite dendrite. In the heat-affected zone, the large fraction of the residual austenite and free carbon resulted in a lower hardness compared to the M1. On enhancing the depth from the M1, the microhardness continuously decreased as the fraction of the residual austenite increased, while the fraction of the ledeburite dendrites and needle-like cementite reduced. Finally, with the depth extending to the substrate, the microhardness decreased sharply.

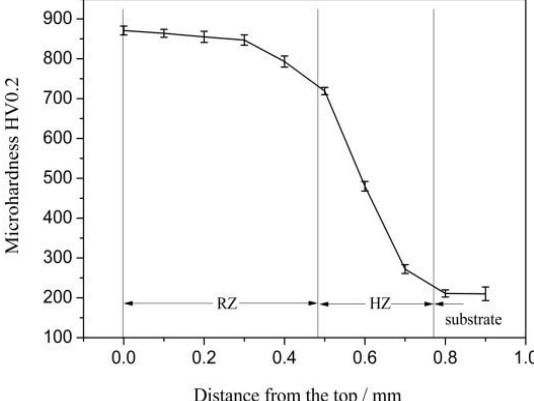

**Figure 6.** Microhardness distribution along the cross-section of the specimens.

It can be seen that the hardness of the M2 proposed in this paper is about 581 HV0.2, which is close to the micro-hardness of the working layer of rolls manufactured by centrifugal casting [27]. In addition, an adequate thickness (approximate 0.2 mm) between the M2 position and the substrate was determined to ensure the safety of wear-resistance workpieces. Therefore, the method of using two-thirds as the surface repair time proposed in this paper was feasible and met the performance requirements of wear-resistant workpieces.

### 3.2. X-Ray Diffraction Analysis

An XRD analysis was carried out to confirm the phase composition of the M1 and M2, as shown in Figure 7. It can be seen that the M1 and M2 consisted of mainly graphite, $\alpha$-Fe, and cementite. It can be confirmed that the microstructure in the remelted gradient layer was in good agreement with the iron-carbon diagram.

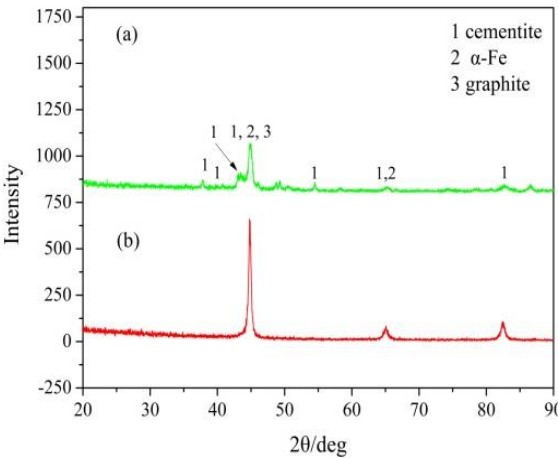

**Figure 7.** XRD diffraction patterns of (a) the the top of the gradient remelted layer (M1) and (b) the position where the hardness was two-thirds of the top of the remelting layer (M2).

### 3.3. Wear Rate and Friction Coefficient

The wear rates of the M1 and M2 at room temperature, determined through the worn surface profiles, are shown in Figure 8. As can be seen, the wear rates of the M1 and M2 were $2.7778 \times 10^{-7}$ and $1.3689 \times 10^{-7}$ g·N$^{-1}$·m$^{-1}$ respectively. Thus, the wear rate of the M2 was about half that of the M1.

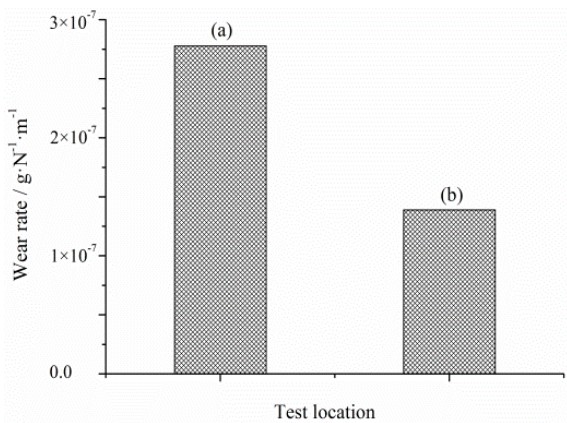

**Figure 8.** The wear rates of (a) the top of the gradient remelted layer (M1), and (b) the position where the hardness was two-thirds of the top of the remelting layer (M2).

The COFs of the M1 and M2 are shown in Figure 9. It can be seen that the COF of the M1 first decreased, followed by an increase to a steady value of approximately 0.22. On the other hand, the COF of the M2 first decreased for a short period of time and subsequently approached a constant value of approximately 0.077. The reason for the lower COF value of the M2 compared to the M1 was the presence of a large amount of carbon atoms in free form serving as lubricant, thus reducing the COF. As the traces of the grinding balls appeared from the M2 to the substrate, the COF became unsteady and continuously increased to higher values due to the variable performance of the different zones. The transformation law of hardness, wear resistance, and coefficient of friction was similar to that described in [23], i.e., the hardness and wear resistance were higher while the coefficient of friction became larger.

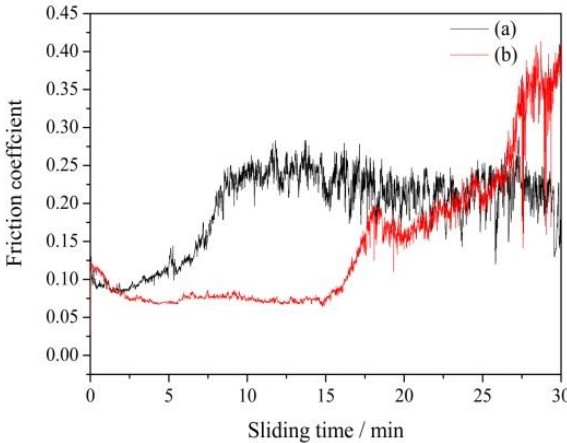

**Figure 9.** Friction coefficient vs. sliding time of (a) the top of the gradient remelted layer (M1) and (b) the position where the hardness was two-thirds of the top of the remelting layer (M2).

*3.4. Worn Surface Morphology*

The worn surfaces and EDS of the M1 and M2 positions are shown in Figure 10 and Table 1. Specifically, Figure 10a,c demonstrates the worn surface of the M1, and Figure 10b,d presents the worn surface of the M2. It can be seen from Figure 10a,b that the worn surfaces in both specimens were relatively smooth and demonstrated oxidative wear, adhesive wear [28], delamination wear [19], and/or fretting wear [29]. In addition, the worn surface of the M2 position was smoother compared to that of the M1 position. This can be attributed to the presence of the soft phase in this region. Furthermore, some cracks and fragments were found to have appeared on the worn surface of the M1 position, while the fragments were peeled on the worn surface of the M2 position. The observed features were induced by many factors, including the high hardness and low plasticity of the M1 position and the ductile iron of the M2 had better toughness than that of the M1 position. In addition, the free carbon in the M2 position acted as a void defect, which reduced the bonding force between the M2 position and the substrate. Therefore, the hardness of the M1 position was much higher than that of the M2 position; however, the wear rate of the M1 was faster than that of the M2 position.

**Table 1.** Results of EDS analysis at different positions according to Figure 10c,d.

| Point Number | Element (at %) | | | | |
|:---:|:---:|:---:|:---:|:---:|:---:|
| | C | Fe | O | Si | Mn |
| 1 | 32.64 | 18.54 | 45.42 | 3.38 | 0.03 |
| 2 | 17.02 | 13.75 | 57.78 | 11.36 | 0.08 |
| 3 | 32.37 | 56.78 | 7.75 | 2.74 | 0.36 |
| 4 | 22.24 | 48.48 | 22.55 | 4.21 | 0.18 |
| 5 | 20.26 | 21.57 | 57.27 | 0.72 | 0.19 |

Furthermore, particles appeared on the worn surfaces of the M1 and M2 position, and the O content was high, as shown point 2 and 5 in Figure 10. These were iron oxides, as confirmed by the results of the EDS. In addition, the Si content of point 2 was higher than that of point 5. Furthermore, the silicon element transferred from the $Si_3N_4$ ceramic ball was induced by the high micro-hardness of the M1 position. However, points 1, 3 and 4 included the amounts of C, Fe, and O, as well as the phases of graphite and iron oxides formed, determined according to the Ellingham diagram for selected oxides [30] and Figure 6. It can therefore be further proven that oxidation wear occurs during friction and wear experiments.

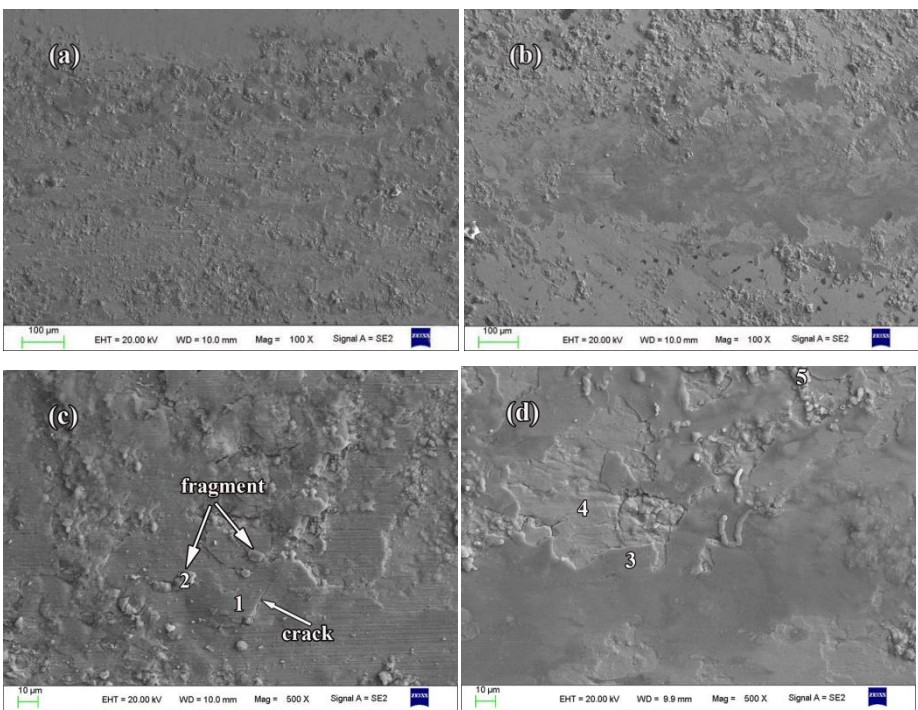

**Figure 10.** SEM micrographs of the worn surfaces of the (**a**,**c**) the top of the gradient remelted layer (M1) and the (**b**,**d**) the position where the hardness was two-thirds of the top of the remelting layer (M2).

## 4. Conclusions

(1) The remelting of the ductile iron was carried out by a plasma transferred arc with arc current of 60 A and a scanning speed of 1000 mm/min. As a result, the M1 consisted mainly of γ-Fe and cementite, and the M2 contained primarily α-Fe, graphite, and cementite.

(2) Although the hardness of the M1 was much higher than that of the M2, the method of using two-thirds as the surface repair time proposed in this paper was feasible. In addition, the wear rate of the M1 was faster than that of the M2. This can be attributed to the presence of free carbon in the M2. The COF of the M1 was noted to be first higher and subsequently lower than that of the M2, due to the transformation of the microstructure from the soft phase with α-Fe to the hard phase with ledeburite dendrites and cementite. The abrasion mechanisms of both regions were complex, which included oxidative wear, adhesive wear, delamination wear, and/or fretting wear.

(3) Cracks and fragments were present on the worn surface of the remelted gradient layer. These features were induced by many factors, especially the free carbon in the M2 acting as a void defect, which reduced the bonding force between the M1 and the substrate. However, a small crack and fragment were noted on the worn surface of the M2, which can be attributed to the presence of a large amount of α-Fe in the substrate.

**Author Contributions:** Investigation, X.Y.; methodology, B.X. and Q.H.; formal analysis, J.F.; data curation, B.X. and J.X.; writing—original draft, B.X.; review and editing, W.J. and Z.F.; funding acquisition, B.X. All authors have read and agreed to the published version of the manuscript.

**Funding:** This research is funded by National Nature Science Foundation of China (no. 51865014) and the State Key Laboratory of Materials Processing and Die & Mould Technology, Huazhong University of Science and Technology (no. P2019-012).

**Institutional Review Board Statement:** Not applicable.

**Informed Consent Statement:** Not applicable.

**Data Availability Statement:** The data presented in this study are available upon request from the corresponding authors.

**Conflicts of Interest:** The authors declare no conflict of interest.

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
