# Peer review of "Comparative Study on the Hardness and Wear Resistance of the Remelted Gradient Layer on Ductile Iron Fabricated by Plasma Transferred Arc"

_metals, doi:10.3390/met12040644_

Round 1

Reviewer 1 Report

This paper is devoted to the study of the hardness and wear resistance of the surface of ductile iron treated with a plasma beam. In general, the subject of the manuscript is interesting and significant for specialists in the field of ferrous metals production. The introduction reflects previously conducted research, the conclusion describes in detail the results of the study. However, there are some comments that need to be corrected.
1. It is not clear which specific areas correspond to M1 and M2. Please provide an image (or diagram) indicating the M1 and M2 areas.
2. Figure 3. Provide a transcript of RZ and HZ in the Figure caption.
3. Figure 4. Martensite and cementite look the same. It may require a large magnification of the SEM image.
4. Figure 5 shows an interesting effect. However, I propose to present Figure 5 in high resolution and indicate the measurement error for each point.
5. Figure 6 should also be presented in high resolution.
6. The same applies to Figures 7 and 8.
7. The Center of the Figure 9d. What's it? Isn't that a crack?
8. I also recommend the authors to provide the scheme of "plasma beam surface modification processing center".

Author Response

Dear Reviewer,

Thank you for your comment, according your recommendation, our manuscript was modified. Please see the attachment.

Yours sincerely,

Botao Xiao

Reviewer 2 Report

This is the comments on the Manuscript Number: Manuscript ID: metals-1640485
Type of manuscript: Article
Title: Comparative Study on Hardness and Wear Resistance of Gradient Remelted
Layer on the Ductile Iron Fabricated by Plasma Beam
Authors: Botao Xiao *, Xuefang Yan, Wenming Jiang, Zitian Fan, Qiwen Huang,
Jun Fang, Junhuai XiangRate the Manuscript:

  1. Significance to field and specialization of “Metals” journal: good.

The paper contains the Surface repair of wear resistant parts is a main application field
of surface treatment, but the repair time of the wear resistant parts is
often not concerned. In fact, the repair time of the wear-resisting parts is
very important, which affects the repair quality and service life of the
wear-resisting parts. So the repair time of the wear-resisting parts, two
thirds of the surface hardness of the wear-resisting parts, was put forward
in this paper, and this method was verified by comparative study on
microhardness and wear resistance of gradient remelted layer on the ductile
iron fabricated by plasma beam. The results show that the microhardness of
the surface of the gradient remelted layer (M1) was higher than that of the
area whose microhardness was the surface hardness of two thirds (M2, ~ 0.5 mm
depth from the top of the surface of the gradient remelted layer), however,
the wear resistance of the M1 was worse, as confirmed by the wear rates. In
addition, the cracks and fragments were observed on the worn surface of the
M1 and M2, while no such cracks were noted on the surface of the M2. Also,
the coefficient of friction (COF) of the M1 is noted to be first higher and
subsequently lower than that of the M2, owing to the grinding ball entering
the substrate. The abrasion mechanisms of both regions were observed to be
complex, including adhesive wear, delamination wear and/or fretting wear.
Experimental data indicate that it is feasible to determine the repair time
according to the microhardness of the parts.2. Scientific content:   good.

  1. Originality: good.
  2. Clarity and presentation: acceptable.
  3. Appropriateness for Journal: appropriate subject mater for the “Metals”
  4. Need for rapid publication: no.

What is the main question addressed by the research?
Do the topic original or relevant to the field? Does it
address a specific gap in the field? -Yes.

The remelting of the ductile iron was carried out by the plasma beam with arc current of 60 A and scanning speed of 1000 mm/min. As a result, the M1 consisted of mainly γ-Fe, martensite and cementite, and the M2 contained primarily α-Fe, graphite and cementite.

Although the hardness of the M1 was much higher than that of the M2, the method of using two thirds as the surface repair time proposed in this paper was feasible. In addition, the wear rate of the M1 was faster than that of the M2. It can be attributed to the present of the free carbon in the M2. The COF of the M1 was noted to be first higher and subsequently lower than that of the M2, due to the transformation of the micro- structure from the soft phase with α-Fe to the hard phase with ledeburite dendrites and  martensite. 3. What does it add to the subject area compared with other published
material?

For example: Laser treatment of plasma coatings // Soviet Materials Science. - New York: Plenum Publishing Corporation.- 1991, vol.27, No 1.- P.51-55. https://doi.org/10.1007/BF00724136

Evaluation of hydrogen containing gasses losses during wear of piston engine // Materials Science. -  2017. – Vol.53, N 2. – P. 156 -159. https://doi.org/10.1007/s11003-017-0074-y

The abrasion mechanisms of both regions were complex, which included adhesive wear, delamination wear and/or fretting wear. The cracks and fragments were present on the worn surface of the gradient re- melted layer. These features were induced by many factors, especially the free carbon in the M2 acting as a void defect, which reduces the bonding force between the M1 and substrate. However, a little crack and fragment were noted on the worn surface of the M2, which can be attributed to the presence of an abundant amount of the α-Fe in the substrate.

What specific improvements should the authors consider regarding the
methodology? What further controls should be considered?

OK
Are the conclusions consistent with the evidence and arguments
presented and do they address the main question posed?

Mainly- yes.
Are the references appropriate?

Mainly- yes.
7. Additional comments on the figures:7, 8 must be improved.

Author Response

Dear Reviewer,

Thank you for your comment on our paper. Thank you for your recognition of research significance of our work. According your recommendation, our manuscript was modified as follows.

Point 1: What does it add to the subject area compared with other published material?

Response 1: Thank you for your comments. You are right, and the subject area of this paper should be added. The method proposed in this paper is mainly for roll repair, and the abstract part is added in the revised manuscript according to the literatures you recommended. It was revised as “Repairing the worn surface of wear resistant workpieces, such as roller, is a main application field of surface treatment, but the repair time of the wear resistant workpieces is often not concerned.”

Yours sincerely,

Botao Xiao

Reviewer 3 Report

Dear Authors,

Please find some comments below.

Abstract

“So the repair time of the wear-resisting parts, two thirds of the surface hardness of the wear-resisting parts, was put forward in this paper...”

Comment: Incomprehensible sentence

“The  results  show  that  the  microhardness  of  the  surface  of  the gradient remelted layer (M1) was higher than that of the area whose microhardness was the surface hardness of two thirds ...”

Comment: It is also confusing and not clear.

“In addition, the cracks and fragments were observed on the worn surface of the...”

Comment: There were observed fragments of what?

“Also, the coefficient of friction (COF) of the M1 is noted to be first higher and subsequently lower than that of the M2,..”

Comment: To make the sentence clear, you must apply to grammar rules.

“...owing to the grinding ball entering the substrate.”

Comment: This fragment is completely incomprehensible. What grinding ball was used? How the grinding ball was entering the substrate? What do you mean “entering” the substrate?

Introduction

“So the surface repair of these parts are carried out in  the  production,...”

Comment: What do you mean “in  the  production”?

You still refer to repair, but it is not clear if you mean repair of worn surfaces or repair of defects during the production stage.

“The ductile iron usually serves as a parts...”

Comment: The above is very confusing. The ductile iron is the material not any part.

“Many literature reports indicate that the microstructure and properties of  the  ductile iron were improved using an high energy density beam with the  laser and plasma.”

Comment: The above sentence is wrong, because plasma is not the “beam” heat source.

“Li et al. [17] simulated the remanufacture processing of  the  ductile iron using laser cladding to improve the  microstructure  and  mechanical  properties  by  different  groove  and  cross  cladding process, the results indicate the microstructure of  the  surface cladding is  the  martensite and the  lamellar cementite, and the microhardness decreased significantly in the bottom of the grooves and the ultimate strength of  the  specimen was 502 MPa.”?

Comment: What do you mean “remanufacture processing”? What do you mean “different  groove” The “grove” is not a typical parameter of laser cladding. What is the “cross  cladding process”?

“Cheng et al. [18] comparative analyzed the surface melted microstructure and properties of the gray and nodular cast irons by  the  plasma beam...”

Comment: The above indicates that “melted microstructure and properties” were analysed. The grammar and English in general must be deeply verified. The term “plasma beam” is not correct. You must verify the terminology and use the proper technical and scientific terminology.

 “At the same time, the microhardness  in  molten  surface  layer...”

Comment: Logical contradiction – molten metal is just liquid, so hardness cannot be determined under such conditions.

“...results show a lower case depth  and  a  surface hardness were measured in the ductile iron treated by the lower energy intensity laser while the wear resistance of the ductile iron was enhanced.”

Comment: What is the “case depth”?

“...ductile iron by  the  plasma beam alloying.”

Comment: There is no any “plasma beam alloying”.

“...the surface hardness of two thirds..”

Comment: What does the above mean?

“In this study, the feasibility of this method will...”

Comment: What method do you mean?

Summarizing,

The English and terminology are confusing. The described process is simply plasma surface melting. The range of the study and results do not exceed what can be found in the literature. The range of the tests is typical for basic technical report. So, in my opinion the manuscript does not meet the requirements for publishing in repayable journal.

Author Response

Dear Reviewer; 

Thank you for your comment, according your recommendation, our manuscript was modified. Please see the attachment. 

Yours sincerely,

Botao Xiao

Reviewer 4 Report

Manuscript No.: Metals, 1640485_2022

Date received: March 07, 2022

Title: Comparative Study on Hardness and Wear Resistance of Gradient Remelted Layer on the Ductile Iron Fabricated by Plasma Beam

Authors: Botao Xiao, Xuefang Yan, Wenming Jiang, Zitian Fan, Qiwen Huang, Jun Fang and Junhuai Xiang.

According to the Abstract the paper refers to the repair time of wear-resistant parts, and the method applied in this regard was verified by a comparative study on the micro-hardness and wear resistance of the layer fused in gradient on ductile iron made of plasma beam.

After carefully reviewing this paper, I recommend that it:

Abstract:

Line 22-23 The authors wrote “In addition, the cracks and fragments were observed on the worn surface of the M1 and M2, while no such cracks were noted on the surface of the M2.” a very confusing statement????

In the introductory part you referred to other authors who have this topic as an interest, but you presented only general aspects. Please provide effective, concrete data on the microstructure and mechanical properties obtained by other authors you cite and with whom you wish to compare your work.

On line 81 you introduce like a base material QT500-7, please offer a worldwide equivalents of grade QT500-7 for instance cast iron QT500-7 and its European equivalent EN-GJS-500-7.

In line 108-109 you introduce “The disc was made of the M1 and M2, while the counterpart ball 108 with a diameter of 6.3 mm was composed of Si3N4 ceramic.” Question: why did you use a ceramic ball as a semi-coupler for the tribometer and not a high hardness alloy steel ball bearing?

Line 113 – suggestion “was observed using SEM.” - was observed using scanning electron microscope (SEM).

Regarding figure 1 - please position the figures as close as possible to the place where you first refer to the text.

Line 146 - FISEM images please explain the abbreviation.

General remarks: The paper combines interesting the structural a mechanical properties aspect of the gradient remelted layer on the ductile iron fabricated by plasma beam and I consider that none of methods can be considered original, but the motivations and goals of the experimental efforts are to be considered to the reader. Nonetheless, the paper can be of interest for the audience of Metals, only after major improvements to this work.

Author Response

Thank you for your comment, according your recommendation, our manuscript was modified as follows.

Point 1: Line 22-23 The authors wrote “In addition, the cracks and fragments were observed on the worn surface of the M1 and M2, while no such cracks were noted on the surface of the M2.” a very confusing statement?

Response 1: Sorry reviewer and editor. “In addition, the cracks and fragments were observed on the worn surface of the M1 and M2, while no such cracks were noted on the surface of the M2.” is a very confusing. I want to say that worn surface M1 has a lot of cracks, and M2 has a small number of cracks. This discussion has been modified in the revised manuscript.

Point 2: In the introductory part you referred to other authors who have this topic as an interest, but you presented only general aspects. Please provide effective, concrete data on the microstructure and mechanical properties obtained by other authors you cite and with whom you wish to compare your work.

Response 2: Thank you for your comments. According to your suggestion, we have revised part of the introduction in the revised manuscript.

Point 3: On line 81 you introduce like a base material QT500-7, please offer a worldwide equivalents of grade QT500-7 for instance cast iron QT500-7 and its European equivalent EN-GJS-500-7.

Response 3: Thank you for your comments. We checked the worldwide equivalents of grade QT500-7. There are many equivalents, such as European standards, American standards, and so on. we are really not sure about the worldwide equivalents of grade QT500-7. In the end, we chose the European equivalent EN-GJS-500-7 you recommended.

Point 4: In line 108-109 you introduce “The disc was made of the M1 and M2, while the counterpart ball 108 with a diameter of 6.3 mm was composed of Si3N4 ceramic.” Question: why did you use a ceramic ball as a semi-coupler for the tribometer and not a high hardness alloy steel ball bearing?

Response 4: Thank you for your comments. We selected the Si3N4 ceramic served as the counterpart ball because the Si3N4 ceramic ball have the high hardness and thermal stability when the friction and wear experiment. Another reason to choose the Si3N4 ceramic ball is to reduce the influence of counterpart ball on the microstructures of the worn surface because the material we do and a high hardness alloy steel ball belong to the iron based alloy.

Point 5: Line 113 – suggestion “was observed using SEM.” - was observed using scanning electron microscope (SEM).

Response 5: Thank you for your suggestion. We adopted your suggestion and modified it in the revised manuscript.

Point 6: Regarding figure 1 - please position the figures as close as possible to the place where you first refer to the text.

Response 6: Thank you for your comment. We put the figures 1as close to the place where we first refer to in the text.

Point 7: Line 146 - FISEM images please explain the abbreviation.

Response 7: Thank you for your comment. FISEM is the abbreviation of the field emission scanning electron microscope. the field emission scanning electron microscope (FISEM) was added to the revised manuscript.

Round 2

Reviewer 1 Report

The authors took into account most of the comments. I believe that the article can be accepted for publication. Additionally, I ask the authors to increase the signatures in Figure 2.

Author Response

Dear reviewer,

 Thank you for your comments. It can help me to improve our manuscript and increase the significance of our manuscript. According to your comments, we modified the signatures in Figure 2. However, it is difficult to modify because of the messages of two types involved in Figure 2. After analysis, we modified the signatures of Figure 2 from “(a) a diagram of plasma processing; (b) the definitions of M1 and M2 in this paper” to “Schematic diagram of working principle of plasma transferred arc and the definitions of M1 and M2 in this paper”.

Yours sincerely,

Botao Xiao

Reviewer 3 Report

The manuscript is deeply modified. I have no more comments and accept it for publication as it is.

Author Response

Dear reviewer,

Thank you for your comments. It can help me to improve our manuscript and increase the significance of our manuscript.

Yours sincerely,

Botao Xiao

Reviewer 4 Report

The authors responded to all suggestions as such I agree with the publication of the paper in this form.

Author Response

(The authors gave the same response as above.)
